# Characterization of Mammary Tumors Arising from MMTV-PyVT Transgenic Mice

Chien-Liang Liu [1], Wen-Chien Huang [1], Shih-Ping Cheng [1,2], Ming-Jen Chen [1,2], Chi-Hsin Lin [3,4], Shao-Chiang Chang [3] and Yuan-Ching Chang [1,*]

1 Department of Surgery, MacKay Memorial Hospital and Mackay Medical College, Taipei 104217, Taiwan; chess@mmh.org.tw (C.-L.L.); wjhuang0.4909@mmh.org.tw (W.-C.H.); disgras@mmh.org.tw (S.-P.C.); mjchen@mmh.org.tw (M.-J.C.)
2 Institute of Biomedical Sciences, Mackay Medical College, New Taipei City 252005, Taiwan
3 Department of Medical Research, MacKay Memorial Hospital, Taipei 104217, Taiwan; lcs2174.b519@mmh.org.tw (C.-H.L.); shao.f109@mmh.org.tw (S.-C.C.)
4 Department of Bioscience Technology, Chung Yuan Christian University, Taoyuan City 320314, Taiwan
* Correspondence: changyc@mmh.org.tw; Tel.: +886-2-2543-3535

**Abstract:** Among genetically engineered mouse models of breast cancer, MMTV-PyVT is a mouse strain in which the oncogenic polyoma virus middle T antigen is driven by the mouse mammary tumor virus promoter. The aim of the present study was to perform morphologic and genetic analyses of mammary tumors arising from MMTV-PyVT mice. To this end, mammary tumors were obtained at 6, 9, 12, and 16 weeks of age for histology and whole-mount analyses. We conducted whole-exome sequencing to identify constitutional and tumor-specific mutations, and genetic variants were identified using the GRCm38/mm10 mouse reference genome. Using hematoxylin and eosin analysis and whole-mount carmine alum staining, we demonstrated the progressive proliferation and invasion of mammary tumors. Frameshift insertions/deletions (indels) were noted in the Muc4. Mammary tumors showed small indels and nonsynonymous single-nucleotide variants but no somatic structural alterations or copy number variations. In summary, we validated MMTV-PyVT transgenic mice as a multistage model for mammary carcinoma development and progression. Our characterization may be used as a reference for guidance in future research.

**Keywords:** breast cancer; MMTV-PyVT mice; whole-exome sequencing

## 1. Introduction

Genetically engineered mouse models have contributed extensively to our understanding of disease processes. To study tumorigenesis in breast cancer, mammary-specific or -selective promoters are commonly used in transgenic mouse models. The most widely used regulatory element for inducing mammary-selective transgene expression is the mouse mammary tumor virus (MMTV) long terminal repeat and the promoter of the whey acidic protein (*Wap*), which encodes the milk serum protein [1]. Various strategies involving the loss of tumor suppressor genes or gain of function in oncogenes such as *Erbb2*, *Myc*, *Ccnd1*, polyoma virus middle T (PyVT), and *Hras* have been used to investigate the initiation and progression of breast cancer. Polyoma virus, such as SV40, is a DNA tumor virus that contains a potent transforming protein, and it was demonstrated that the middle T antigen is required for transformation [2]. The advantages of MMTV-PyVT transgenic mice are their development of synchronous multifocal tumors in all of the mammary glands with a short latency, as well as a high prevalence of pulmonary metastasis [3]. As such, MMTV-PyVT is the most commonly used genetically engineered mouse model for cancer research and serves as a preclinical platform for therapeutic testing [4].

Although MMTV-PyVT transgenic mice are frequently used in preclinical research, few studies have performed morphologic and genetic analyses of mammary tumors arising

from MMTV-PyVT mice. Comprehensive genomic profiling revealed that MMTV-PyVT tumors showed luminal-like gene expression patterns [5,6]. Another study evaluated the phenotypes of multiple immunohistochemical markers and demonstrated that Ki-67 expression progressively increased during tumor progression [7]. In the current study, we examined histological characteristics, including whole-mount carmine alum staining, and determined genomic alterations using whole-exome sequencing (WES). The results of this study may shed light on the pathogenesis of the development of breast neoplasms in this model.

## 2. Materials and Methods

### 2.1. Transgenic Mice

This study was conducted in accordance with the Guidelines for the Care and Use of Laboratory Animals published by the Council of Agriculture of Taiwan and was approved by the Institutional Animal Care and Use Committee of MacKay Memorial Hospital (MMH-A-S-108-22). B6.FVB-Tg(MMTV-PyVT)634Mul/LellJ transgenic mice from the Jackson Laboratory (Bar Harbor, ME, USA) represent a mouse strain in which an oncogene derived from the polyoma virus is expressed in the mammary gland tissues, driven by the mammary tumor virus promoter [8]. To maintain a live colony, hemizygous male B6.FVB-Tg(MMTV-PyVT)634Mul/LellJ mice were bred with C57BL/6J inbred females (purchased from BioLASCO, Taipei, Taiwan). Dr. Ming-Shen Dai (Tri-Service General Hospital, Taipei, Taiwan) generously provided the MMTV-PyVT mice [9].

### 2.2. Spontaneous Mammary Tumor

Hemizygous MMTV-PyVT mice develop spontaneous mammary tumors that closely resemble the progression of human breast cancer from premalignant to malignant breast disease. Both the body weight and tumor volume of each mouse were monitored twice a week for the duration of the study. Tumor volume was calculated using the modified ellipsoidal formula as length $\times$ width$^2$ $\times$ 1/2 [10]. Mammary glands and mammary tumors were collected from at least five mice per time point for histological analysis. Normal mammary gland tissue samples were obtained from age-matched C57BL/6J female mice.

### 2.3. Histology and Whole-Mount Analysis

After sacrifice of the mouse, thoracic (second and third) and abdominal (fourth) mammary glands and tumors were harvested and fixed in 10% formalin at room temperature overnight. The tissues were paraffin-embedded, sectioned, and stained as per the standard hematoxylin and eosin (H&E) procedures. To prepare the whole mounts, the mammary glands were transferred onto a positively charged microscope slide (Muto Pure Chemicals, Tokyo, Japan). The mammary glands were spread out as much as possible without tearing the tissue and were fixed with Carnoy's solution composed of 60% ethanol, 30% chloroform, and 10% glacial acetic acid for 24 h. The glands were then stained with carmine alum (Sigma-Aldrich, St. Louis, MO, USA) for 48 h, de-stained in 70% ethanol with 2 mM HCl for 4 h to remove excessive dye, and dehydrated in graded alcohol solution [11]. The dehydrated tissue slides were submerged in xylene for at least 12 h for adipose tissue clearing.

### 2.4. Whole-Exome Sequencing

DNA from mammary tumors and matched tails were extracted from two MMTV-PyVT mice at 12 weeks of age using the DNeasy Blood and Tissue Kits (Qiagen, Hilden, Germany). The extracted DNA was treated with RNase, purified using the QIAamp DNA Micro Kit (Qiagen), and sheared into fragments. Exon capture was performed with the SureSelect XT Mouse All Exon Kit (Agilent, Santa Clara, CA, USA). Exon capture libraries were sequenced using a paired-end protocol on the Illumina NovaSeq 6000 platform (Illumina, San Diego, CA, USA). The sequencing data were deposited in the BioProject database of the National Center for Biotechnology Information with the accession number PRJNA890699.

### 2.5. Exome Data Analysis

The sequencing reads were trimmed by removing low-quality bases and aligned to the mouse reference genome (GRCm38/mm10). Duplicate removal and base quality recalibration were performed using the Genome Analysis Toolkit (Broad Institute). The 20× mean depth coverage rate for the samples was 85.86 ± 1.12%. On average, 45.48 ± 3.01 M mapped reads with a duplication rate of 22.65 ± 1.65% were obtained. Structural variants were called using Manta v.1.6.0 [12] and were annotated using AnnotSV v2.2 [13]. For copy number variant analysis, segments were filtered for significance using the following criteria: Wilcoxon's rank sum test with a *p*-value < 0.001, the Kolmogorov–Smirnov test with a *p*-value < 0.001, and uncertainty between 0 and 20. For insertion/deletion (indel) and single-nucleotide variant (SNV) detection, genetic variants were annotated with ANNOVAR [14], and the effect of each variant on the coding sequences was predicted using SnpEff v5.1 [15].

### 3. Results

#### 3.1. Spontaneous Tumor Formation

All MMTV-PyVT mice develop spontaneous tumors arising from the mammary pads within a predictable time frame (Figure 1). Tumor formation was generally visible at an average of 11 weeks of age. We chose four time points corresponding to breast tumor formation for histology and whole-mount analysis: week 6 for hyperplasia, week 9 for ductal carcinoma in situ (DCIS), week 12 for early carcinoma, and week 16 for late carcinoma.

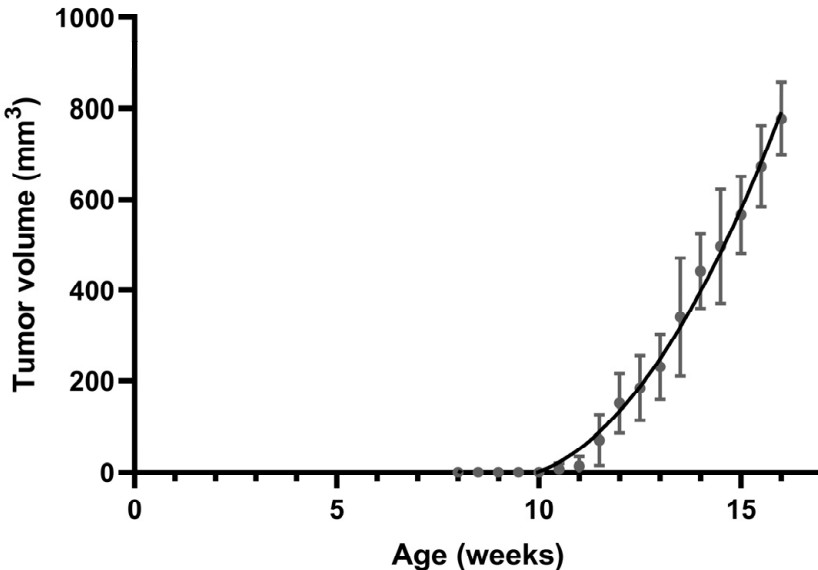

**Figure 1.** Spontaneous mammary tumor growth in female MMTV-PyVT mice. Data are shown as mean ± SD (*n* = 7).

#### 3.2. Histological Analysis

At 6 weeks of age, low-grade proliferation of the polarized glandular epithelial cells was evident on the H&E-stained sections of mammary glands (Figure 2). Most cells had a columnar shape with a low mitotic index. At 9 weeks, hyperplastic cells were haphazardly arranged along the duct wall of the terminal duct lobular unit. The cell borders were indistinct, and some neoplastic cells were loosely adhered to the duct walls. The neoplastic proliferation was confined within the lumens of the involved ducts and lobules. At 12 weeks of age, neoplastic cells with mild to moderate atypia breached the basement membrane around the lobular glands. Reactive alterations were present in the surrounding stroma. At 16 weeks, apparent features of late-stage carcinoma were accompanied by a grossly palpable lesion. The fibrotic tumor stroma was dense and collagenous, and angiolymphatic space invasion was sparsely identified.

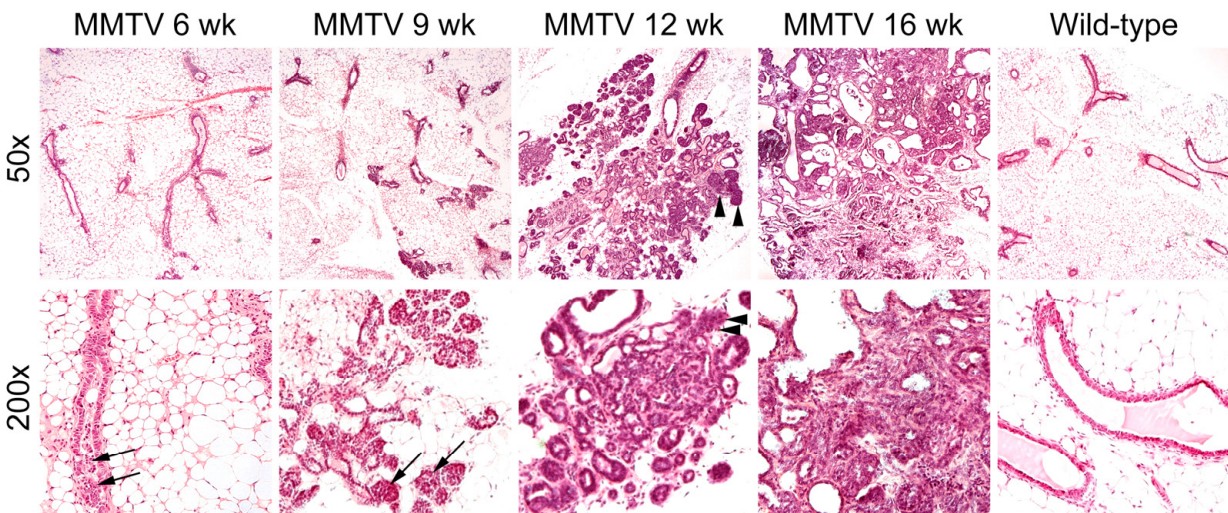

**Figure 2.** Hematoxylin and eosin staining of mammary glands in female wild-type and MMTV-PyVT transgenic mice. Original magnification: upper panel, ×50; lower panel, ×200. Arrows, hyperplastic intraductal cells; arrowheads, neoplastic growth through the basement membrane.

Thoracic and abdominal mammary fat pads containing entire mammary glands were placed on whole-mount slides and stained with carmine alum to depict the epithelial structures of the mammary gland. A higher branching density and increased staining intensity in the bulb-shaped terminal end buds were observed as early as 6 weeks of age (Figure 3). Along with tumor progression, there was an increase in the number and extent of hyperplastic areas.

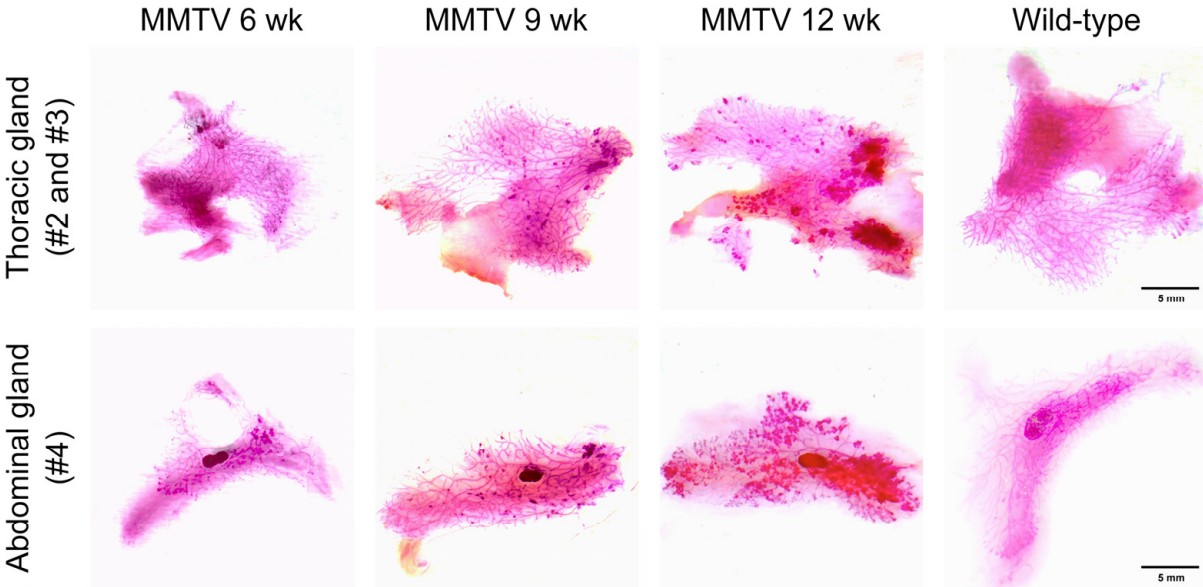

**Figure 3.** Carmine-alum-stained thoracic (second and third) and abdominal (fourth) mammary glands of female wild-type and MMTV-PyVT transgenic mice. Scale bar, 5 mm.

### 3.3. Mouse-Level Genetic Alterations

We performed WES on mammary tumor and tail genomic DNA to identify structural variants and mutations in the MMTV-PyVT transgenic mice. Genetic variants in the WES data were identified using the mm10 mouse reference genome, and only variants present in all four samples were included. As shown in Table 1, the majority of variants were

deletions, with the lengths of the variants ranging from 51 to 57,900 bp. Small indels are listed in Table 2. It is noteworthy that *Muc4*, an oncogene in tier-2 genes in the Cancer Gene Census, had frameshift indels on chromosome 16. SNVs with nonsynonymous mutations are listed in Table 3. We did not find nonsynonymous SNVs among the genes in the Cancer Gene Census [16].

**Table 1.** Structural variants of MMTV-PyVT transgenic mice.

| Location | Gene | Accession No. | Class |
|---|---|---|---|
| Chr1:88234242-88234312 | *Mroh2a* | NM_001281466 | DEL, INV |
| Chr1:88266278-88266329 | *6430706D22Rik* | NR_040291 | DEL |
| Chr1:88270315-88271447 | *A730008H23Rik* | NM_172505 | DEL |
| Chr1:88270315-88271447 | *Hjurp* | NM_198652 | DEL |
| Chr1:171345007-171356531 | *Nit1* | NM_012049 | DEL, BND |
| Chr1:171356616-171356702 | *Pfdn2* | NM_001360825 | DEL |
| Chr6:116022400-116024804 | *Tmcc1* | NM_001364577 | DEL |
| Chr7:24082235-24085425 | *Zfp180* | NM_001045486 | DEL |
| Chr7:79651636-79665662 | *Ticrr* | NM_029835 | DEL |
| Chr7:84947480-84947640 | *Vmn2r65* | NM_001105180 | DEL |
| Chr7:141638851-141639059 | *Muc6* | NM_001368953 | DEL |
| Chr8:35482726-35491202 | *Eri1* | NM_026067 | DEL |
| Chr9:56260762-56318663 | *Peak1* | NM_172924 | DEL |
| Chr10:81178233-81178679 | *Eef2* | NM_007907 | DEL |
| Chr11:3137135-3137136 | *Sfi1* | NM_030207 | INS, BND |
| Chr12:106051004-106051759 | *Vrk1* | NM_001029843 | DEL |
| Chr19:21598332-21598752 | *1110059E24Rik* | NM_025423 | DEL |

Abbreviation: DEL, deletion; INS, insertion; INV, inversion; BND, break-end.

**Table 2.** Insertion/deletion (Indel) mutations of MMTV-PyVT transgenic mice.

| Location | Gene | Accession No. | Indel | Type |
|---|---|---|---|---|
| Chr1:139237087-139237087 | *Crb1* | NM_133239 | c.3481delC, p.R1161Gfs*48 | frameshift deletion |
| Chr1:85094333-85094335 | *A530032D15Rik* | NM_213615 | c.392_394del, p.A131del | non-frameshift deletion |
| Chr1:85591567-85591567 | *Sp110* | NM_030194 | c.538dupG, p.A180Gfs*26 | frameshift insertion |
| Chr1:88212372-88212372 | *Ugt1a1* | NM_201645 | c.371delT, p.M124Sfs*3 | frameshift deletion |
| Chr1:88216161-88216162 | *Ugt1a2* | NM_013701 | c.1108_1109del, p.I370Hfs*10 | frameshift deletion |
| Chr1:9546104-9546105 | *Rrs1* | NM_021511 | c.581_582del, p.H197Qfs*18 | frameshift deletion |
| Chr3:130040795-130040797 | *Sec24b* | NM_207209 | c.751_753del, p.Q251del | non-frameshift deletion |
| Chr3:96683463-96683463 | *Ankrd35* | NM_001081139 | c.1065dupA, p.N356Kfs*9 | frameshift insertion |
| Chr4:63171423-63171423 | *Kif12* | NM_010616 | c.179_180insGCCGGGTGGAGGCCC, p.P60_D61insPGGGP | non-frameshift insertion |
| Chr5:32737988-32737988 | *Pisd* | NM_177298 | c.C994T, p.R332X | stopgain |
| Chr5:93637618-93637618 | *Pramel34* | NM_001164284 | c.C802T, p.Q268X | stopgain |
| Chr8:104182034-104182034 | *Bean1* | NM_001141922 | c.42delC, p.Q15Kfs*29 | frameshift deletion |
| Chr8:26160857-26160858 | *Thap1* | NM_199042 | c.154_155del, p.S52Hfs*12 | frameshift deletion |

**Table 2.** *Cont.*

| Location | Gene | Accession No. | Indel | Type |
|---|---|---|---|---|
| Chr9:103355194-103355204 | *Cdv3* | NM_175833 | c.805_815del, p.V269Sfs*16 | frameshift deletion |
| Chr9:39484333-39484333 | *Or8g20* | NM_146830 | c.919delA, p.I307* | stopgain |
| Chr9:65280131-65280131 | *Cilp* | NM_173385 | c.3507delG, p.G1170Afs*16 | frameshift deletion |
| Chr13:64921972-64921972 | *Spata31* | NM_030047 | c.C1933T, p.R645X | stopgain |
| Chr16:32752550-32752550 | *Muc4* | NM_080457 | c.2427_2428insCA, p.Q810Hfs*30 | frameshift insertion |
| Chr16:44496429-44496431 | *Boc* | NM_172506 | c.1348_1350del, p.S450del | non-frameshift deletion |
| Chr16:45729926-45729926 | *Abhd10* | NM_001272070 | c.870delC, p.D291Tfs*15 | frameshift deletion |
| Chr17:23291424-23291424 | *Vmn2r114* | NM_001102584 | c.C2081G, p.S694X | stopgain |
| Chr17:23475353-23475353 | *Vmn2r117* | NM_001104581 | c.G1519T, p.G507X | stopgain |

**Table 3.** Nonsynonymous single-nucleotide variants (SNVs) of MMTV-PyVT transgenic mice.

| Location | Gene | Accession No. | SNV |
|---|---|---|---|
| Chr1:12839899-12839899 | *Sulf1* | NM_172294 | c.G1918A, p.D640N |
| Chr1:177773679-177773679 | *Adss* | NM_007422 | c.C686T, p.P229L |
| Chr1:26687400-26687400 | *4931408C20Rik* | NM_001033764 | c.T27A, p.N9K |
| Chr1:59847167-59847167 | *Bmpr2* | NM_007561 | c.G962A, p.R321Q |
| Chr1:75486756-75486756 | *Obsl1* | NM_178884 | c.C5291T, p.T1764M |
| Chr1:85246950-85246950 | *C130026I21Rik* | NM_175219 | c.C863T, p.S288L |
| Chr1:85615212-85615212 | *Sp140* | NM_001013817 | c.C443G, p.T148R |
| Chr2:131936461-131936461 | *Prnp* | NM_011170 | c.T32A, p.L11H |
| Chr2:84872044-84872044 | *Rtn4rl2* | NM_199223 | c.G1165C, p.A389P |
| Chr3:144691910-144691910 | *Sh3glb1* | NM_019464 | c.T980A, p.L327Q |
| Chr3:15548939-15548939 | *Sirpb1b* | NM_001173460 | c.A82G, p.M28V |
| Chr3:95734876-95734876 | *Ecm1* | NM_007899 | c.A1396G, p.I466V |
| Chr3:96854557-96854557 | *Pdzk1* | NM_021517 | c.A484G, p.N162D |
| Chr4:138221673-138221673 | *Hp1bp3* | NM_010470 | c.C46T, p.L16F |
| Chr4:140798123-140798123 | *Padi3* | NM_011060 | c.T578C, p.L193P |
| Chr4:147510785-147510785 | *Zfp982* | NM_001039209 | c.A63C, p.E21D |
| Chr4:147581328-147581328 | *Zfp985* | NM_001014397 | c.T117G, p.I39M |
| Chr4:147613775-147613775 | *Zfp979* | NM_145078 | c.C476T, p.T159I |
| Chr4:148944359-148944359 | *Casz1* | NM_027195 | c.T3260C, p.L1087P |
| Chr4:21873684-21873684 | *Pnisr* | NM_025669 | c.C1426G, p.R476G |
| Chr4:3184971-3184971 | *Vmn1r3* | NM_001167535 | c.C335T, p.T112I |
| Chr5:112762721-112762721 | *Myo18b* | NM_028901 | c.G5804A, p.R1935H |
| Chr5:114398443-114398443 | *Ube3b* | NM_054093 | c.T749G, p.M250R |
| Chr5:13570208-13570208 | *Sema3a* | NM_009152 | c.A1423G, p.I475V |
| Chr5:26035024-26035024 | *Speer4a* | NM_029376 | c.A727C, p.T243P |
| Chr5:27501274-27501274 | *Speer4b* | NM_028561 | c.C94T, p.P32S |
| Chr5:38300085-38300085 | *Otop1* | NM_172709 | c.G1187C, p.G396A |

**Table 3.** *Cont.*

| Location | Gene | Accession No. | SNV |
|---|---|---|---|
| Chr5:89775351-89775351 | *Adamts3* | NM_177872 | c.G595A, p.V199I |
| Chr5:96758142-96758142 | *Fras1* | NM_175473 | c.T9404C, p.L3135P |
| Chr6:39400456-39400456 | *Mkrn1* | NM_018810 | c.A1036T, p.N346Y |
| Chr7:102973309-102973309 | *Or51g2* | NM_147109 | c.G682A, p.V228I |
| Chr7:105434593-105434593 | *Cckbr* | NM_007627 | c.C727G, p.R243G |
| Chr7:108465371-108465371 | *Or5p73* | NM_146307 | c.T46A, p.F16I |
| Chr7:120135179-120135179 | *Zp2* | NM_011775 | c.C1646T, p.A549V |
| Chr7:122167650-122167650 | *Plk1* | NM_011121 | c.C1090T, p.R364W |
| Chr7:131065072-131065072 | *Dmbt1* | NM_007769 | c.C1342A, p.P448T |
| Chr7:13801414-13801414 | *Sult2a1* | NM_001111296 | c.A713G, p.Q238R |
| Chr7:141858623-141858623 | *Muc5b* | NM_028801 | c.T5305C, p.Y1769H |
| Chr7:3222537-3222537 | *Nlrp12* | NM_001033431 | c.A3101G, p.K1034R |
| Chr7:43187290-43187290 | *Zfp936* | NM_001034893 | c.G124A, p.A42T |
| Chr7:56131292-56131292 | *Herc2* | NM_010418 | c.G3704A, p.G1235D |
| Chr7:79111354-79111354 | *Acan* | NM_007424 | c.A5813C, p.H1938P |
| Chr7:92858589-92858589 | *Ddias* | NM_001080995 | c.C2117T, p.P706L |
| Chr8:122890181-122890181 | *Ankrd11* | NM_001081379 | c.G6868C, p.V2290L |
| Chr9:109145537-109145537 | *Fbxw21* | NM_177069 | c.A914G, p.H305R |
| Chr9:120016907-120016907 | *Xirp1* | NM_011724 | c.A2909C, p.Q970P |
| Chr9:25130622-25130622 | *Herpud2* | NM_020586 | c.G253C, p.V85L |
| Chr9:38581513-38581513 | *Or8b48* | NM_146810 | c.C235T, p.P79S |
| Chr9:44249891-44249891 | *Pdzd3* | NM_133226 | c.T472C, p.C158R |
| Chr9:44942695-44942695 | *Ube4a* | NM_145400 | c.A1747T, p.N583Y |
| Chr10:58231344-58231344 | *Dux* | NM_001081954 | c.G1332C, p.L444F |
| Chr10:67238174-67238174 | *Jmjd1c* | NM_001242396 | c.T5144C, p.L1715P |
| Chr10:79169477-79169477 | *Vmn2r80* | NM_001103368 | c.A947G, p.N316S |
| Chr10:88091833-88091833 | *Pmch* | NM_029971 | c.T395C, p.I132T |
| Chr11:46222615-46222615 | *Cyfip2* | NM_133769 | c.C2903T, p.S968F |
| Chr11:90480671-90480671 | *Stxbp4* | NM_011505 | c.G1603A, p.A535T |
| Chr13:100161909-100161909 | *Naip2* | NM_010872 | c.T1618A, p.Y540N |
| Chr13:21468303-21468303 | *Nkapl* | NM_025719 | c.G139C, p.G47R |
| Chr13:27272475-27272475 | *Prl3a1* | NM_025896 | c.C311T, p.T104I |
| Chr13:53117204-53117204 | *Ror2* | NM_013846 | c.G1114A, p.V372M |
| Chr13:93063579-93063579 | *Cmya5* | NM_023821 | c.G10240C, p.A3414P |
| Chr14:51413192-51413192 | *Vmn2r88* | NM_001368932 | c.A361G, p.T121A |
| Chr14:70586204-70586204 | *Fhip2b* | NM_194345 | c.C1725A, p.S575R |
| Chr15:77638007-77638007 | *Apol11b* | NM_001143686 | c.T89G, p.I30R |
| Chr16:35291544-35291544 | *Adcy5* | NM_001012765 | c.G2770A, p.V924M |
| Chr16:36772445-36772445 | *Slc15a2* | NM_021301 | c.T629C, p.M210T |
| Chr16:38828345-38828345 | *Tex55* | NM_029042 | c.C401G, p.T134S |

**Table 3.** *Cont.*

| Location | Gene | Accession No. | SNV |
|---|---|---|---|
| Chr16:39024953-39024953 | *Igsf11* | NM_170599 | c.G1045C, p.A349P |
| Chr16:43939116-43939116 | *Ccdc191* | NM_027801 | c.G1279A, p.V427I |
| Chr16:44299802-44299802 | *Sidt1* | NM_198034 | c.C515G, p.P172R |
| Chr16:44379308-44379308 | *Spice1* | NM_144550 | c.C2122A, p.R708S |
| Chr16:44789572-44789572 | *Cd200r1* | NM_021325 | c.A153G, p.I51M |
| Chr16:44820915-44820915 | *Cd200r4* | NM_207244 | c.T20C, p.I7T |
| Chr16:45094982-45094982 | *Ccdc80* | NM_026439 | c.A100G, p.T34A |
| Chr16:45239239-45239239 | *Btla* | NM_177584 | c.A305G, p.Q102R |
| Chr16:45392332-45392332 | *Cd200* | NM_010818 | c.A751G, p.I251V |
| Chr16:45539592-45539592 | *Slc9c1* | NM_198106 | c.T8C, p.M3T |
| Chr16:45664252-45664252 | *Tmprss7* | NM_172455 | c.G1544T, p.S515I |
| Chr16:46049747-46049747 | *Cd96* | NM_032465 | c.T1358C, p.F453S |
| Chr16:48817255-48817255 | *Retnlb* | NM_023881 | c.C43T, p.L15F |
| Chr17:35425194-35425194 | *H2-Q6* | NM_207648 | c.A151G, p.N51D |
| Chr17:35440154-35440154 | *H2-Q7* | NM_010394 | c.C580G, p.Q194E |
| Chr17:45517174-45517174 | *Aars2* | NM_198608 | c.C1810T, p.R604C |
| Chr17:47400410-47400410 | *Guca1a* | NM_008189 | c.A10G, p.I4V |
| Chr17:67752883-67752883 | *Lama1* | NM_008480 | c.G1966A, p.D656N |
| Chr18:24017781-24017781 | *Zfp24* | NM_021559 | c.A307T, p.I103F |
| ChrX:124127783-124127783 | *Vmn2r121* | NM_001100616 | c.A2539T, p.N847Y |

### 3.4. Tumor-Level Genetic Alterations

The WES data from the mammary tumors were compared with those from matched tails. Genetic variants present in the mammary tumor DNA but not in the tail genomic DNA were considered somatic. There were no somatic structural alterations or copy number variations. Two oncogenes in tier-1 genes in the Cancer Gene Census, *Kat6a* and *Kmt2d*, had non-frameshift deletions. Table 4 lists somatic indels and SNVs of the mammary tumors from the MMTV-PyVT transgenic mice.

**Table 4.** Somatic insertion/deletion and nonsynonymous single-nucleotide variants (SNVs) of mammary tumors arising from MMTV-PyVT transgenic mice.

| Location | Gene | Accession No. | Amino Acid Change | Type |
|---|---|---|---|---|
| Chr3:15411378-15411378 | *Sirpb1a* | NM_001002898 | c.G559A, p.D187N | nonsynonymous SNV |
| Chr5:145803665-145803665 | *Cyp3a44* | NM_177380 | c.C164A, p.T55K | nonsynonymous SNV |
| Chr5:94535811-94535811 | *Pramel42* | NM_001243937 | c.T299A, p.L100H | nonsynonymous SNV |
| Chr6:29441097-29441102 | *Flnc* | NM_001081185 | c.1051_1054del, p.V351Pfs*16 | frameshift deletion |
| Chr7:35409643-35409645 | *Cep89* | NM_028120 | c.546_548del, p.S190del | non-frameshift deletion |
| Chr8:122478985-122478987 | *Ctu2* | NM_153775 | c.546_548del, p.Q190del | non-frameshift deletion |
| Chr8:22935648-22935650 | *Kat6a* | NM_001081149 | c.3208_3210del, p.E1077del | non-frameshift deletion |
| Chr9:99583673-99583675 | *Dbr1* | NM_031403 | c.1303_1305del, p.E444del | non-frameshift deletion |
| Chr12:8728945-8728947 | *Pum2* | NM_030723 | c.1516_1518del, p.Q513del | non-frameshift deletion |
| Chr14:98168891-98168893 | *Dach1* | NM_007826 | c.417_419del, p.S156del | non-frameshift deletion |

**Table 4.** *Cont.*

| Location | Gene | Accession No. | Amino Acid Change | Type |
|---|---|---|---|---|
| Chr15:101433138-101433138 | *Krt87* | NM_001003668 | c.T1226C, p.I409T | nonsynonymous SNV |
| Chr15:98846446-98846448 | *Kmt2d* | NM_001033276 | c.10830_10832del, p.Q3610del | non-frameshift deletion |
| Chr17:23348034-23348034 | *Vmn2r115* | NM_001104579 | c.G1519T, p.G507X | stopgain |
| Chr17:35873852-35873854 | *Ppp1r18* | NM_175242 | c.1678_1680del, p.E570del | non-frameshift deletion |
| Chr17:46412515-46412517 | *Zfp318* | NM_207671 | c.5443_5445del, p.E1823del | non-frameshift deletion |

## 4. Discussion

Mouse models of breast cancer play an important role in the study of disease mechanisms and conduction of in vivo pharmacological testing. However, heterogeneity is quite common between models and within models [17]. Several large-scale analyses have been performed to compare the genetic perturbations of mammary lesions arising from different models or during the process of metastasis [3,5,6,17–20]. Discrepancies are the rule, possibly because multiple aberrations may be acquired early or late in breast tumorigenesis. Any attempt to clarify the pathogenesis may provide a new piece of the puzzle that will allow us to further understand the molecular basis of breast cancer initiation and progression.

Two of the principal signaling pathways that are stimulated by the PyVT are the mitogen-activated protein kinase (MAPK) and phosphatidylinositol 3-kinase (PI3K) cascades [21]. A few genetic alterations in this transgenic model have been reported previously. Src activation has been shown to play a pivotal role in PyVT-induced tumor formation [4,22]. Copy number alterations in key extracellular matrix proteins, including *Col1a1* and *Chad*, were shown to drive metastasis in MMTV-PyVT transgenic mice [19]. In another recent study, the presence of previously unreported recurrent mutations in *Shc1*, as well as recurrent oncogenic mutations in *Kras* and *Ctnnb1*, was a key factor driving metastasis in MMTV-PyVT mice [20]. As in real-world patients, the acquisition of varying aberrations occurs in different mice and in different laboratories.

In this study, we found that *Muc4* had frameshift indels in our MMTV-PyVT mice. MUC4 is a member of the transmembrane mucin family, and aberrant expression has been reported in a variety of carcinomas [23]. Aberrantly expressed MUC4 can act as a ligand for ERBB2, potentiate the phosphorylation of ERBB2, and reduce the binding of anti-ERBB2 antibodies to tumor cell surfaces [24]. Recently, it was demonstrated that Muc4 may facilitate tumor cell survival in circulation and, therefore, metastasis by promoting the association of circulating tumor cells with blood cells [25]. Pulmonary metastasis is commonly observed in MMTV-PyVT transgenic mice. In this regard, the frameshifts in *Muc4* may alter Muc4 expression and function and contribute to the high prevalence of metastasis. An important limitation of the current study is that we did not determine Muc4 expression during breast tumor development and progression. Moreover, it would be intriguing to correlate the expression level of Muc4 with proliferative indexes (such as Ki-67) or the expression of pro- and anti-apoptotic markers in breast tumors.

We demonstrated that mammary tumors arise in MMTV-PyVT mice through a multistage process, as in human breast cancer. However, while MMTV-PyVT transgenic mice are very useful in preclinical research, this genetically engineered mouse model does not recapitulate all aspects of human breast cancer. During the development of human breast cancer, gains in oncogene function or losses of tumor suppressor genes occur in a limited number of cells, whereas transgene effects are found throughout the mammary epithelial cells [26]. Nonetheless, MMTV-PyVT mice provide a versatile platform for studying various facets of breast tumorigenesis.

To summarize, we validated MMTV-PyVT transgenic mice as a multistage model for mammary carcinoma development and progression through histological analysis and whole-mount carmine alum staining. Constitutional and somatic genomic alterations were determined using WES, and possible pathogenic frameshift indels of *Muc4* were

identified. Our characterization may be used as a reference for guidance in future research on MMTV-PyVT transgenic mice.

**Author Contributions:** Conceptualization, C.-L.L., M.-J.C. and Y.-C.C.; methodology, W.-C.H., S.-P.C., M.-J.C., C.-H.L. and S.-C.C.; formal analysis, C.-L.L., W.-C.H., S.-P.C. and Y.-C.C.; data curation, M.-J.C., C.-H.L. and S.-C.C.; writing—original draft preparation, C.-L.L. and Y.-C.C.; writing—review and editing, W.-C.H., S.-P.C., M.-J.C., C.-H.L. and S.-C.C.; funding acquisition, W.-C.H. and Y.-C.C. All authors have read and agreed to the published version of the manuscript.

**Funding:** This research was funded by MacKay Memorial Hospital, grant numbers MMH-E-111-08 and MMH-E-112-08. The funders had no role in the design of the study; in the collection, analyses, or interpretation of data; in the writing of the manuscript; or in the decision to publish the results.

**Institutional Review Board Statement:** The animal study protocol was approved by the Institutional Animal Care and Use Committee of MacKay Memorial Hospital (MMH-A-S-108-22).

**Informed Consent Statement:** Not applicable.

**Data Availability Statement:** The sequencing data of this study can be found in the BioProject database of the National Center for Biotechnology Information with the accession number PR-JNA890699.

**Conflicts of Interest:** The authors declare no conflict of interest.

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
