# Peer review of "Characterization of Mammary Tumors Arising from MMTV-PyVT Transgenic Mice"

_cimb, doi:10.3390/cimb45060286_

Round 1

Reviewer 1 Report

Comments and Suggestions for Authors

Liu et al. cimb-2410691 " Characterization of mammary tumors arising from MMTV-PyVT transgenic mice " is a valuable paper showing progressive proliferation and invasion of mammary tumors by using hematoxylin and eosin analysis and whole-mount carmine alum staining. In addition, the authors showed that Muc4 had frameshift indels in MMTV-PyVT mice, and mammary tumors had small indels and non-synonymous single nucleotide variants but no somatic structural alterations or copy number variations. The reviewer was very interested in this paper and could understand the characterization of mammary tumors in MMTV-PyVT mice. Therefore, there is only one suggestion from the reviewer. In Figure 2, the reviewer could not understand the location of the characteristic structure, such as the columnar shape, lobular gland, or grossly palpable lesion, in these stained results. The reviewer believes the reader's understanding would be enhanced by indicating those structures in this Figure with arrowheads.

Author Response

Liu et al. cimb-2410691 "Characterization of mammary tumors arising from MMTV-PyVT transgenic mice" is a valuable paper showing progressive proliferation and invasion of mammary tumors by using hematoxylin and eosin analysis and whole-mount carmine alum staining. In addition, the authors showed that Muc4 had frameshift indels in MMTV-PyVT mice, and mammary tumors had small indels and non-synonymous single nucleotide variants but no somatic structural alterations or copy number variations. The reviewer was very interested in this paper and could understand the characterization of mammary tumors in MMTV-PyVT mice. Therefore, there is only one suggestion from the reviewer. In Figure 2, the reviewer could not understand the location of the characteristic structure, such as the columnar shape, lobular gland, or grossly palpable lesion, in these stained results. The reviewer believes the reader's understanding would be enhanced by indicating those structures in this Figure with arrowheads.

Response:

We thank the reviewer for pointing this out. We have added markers to the relevant findings.

Reviewer 2 Report

Comments and Suggestions for Authors

Authors Chien-Liang Liu and colleagues describes the Characterization of mammary tumors arising from MMTV- 2 PyVT transgenic mice.  Authors analyzed whole-exome sequencing to identify constitutional and tumor-specific mutations, using the GRCm38/mm10 mouse reference genome. Tumors were collected at different time points for analysis including hematoxylin and eosin and whole-mount carmine alum staining. Authors found Frameshift insertions/deletions (indels) in the Muc4 and small indels and nonsynonymous single nucleotide variants but no somatic structural alterations or copy number variations. Though MMTV- 2 PyVT transgenic mice are used as mammary tumor model from long time, the current work author tries to find some new information by whole-exome sequencing.  

In general, the manuscript is written well, importantly helpful for clinical and scientific understanding. There are few issues making the work less exciting and harnessing the readability. Overall, this manuscript is acceptable with the following minor comments:

For the tumors extracted at different time points mainly the authors did histology and whole-mount analysis. Mainly focus is Whole exome sequencing, and data analysis.

1.       It is not clear at what time point samples were collected for whole exome sequencing.

2.       Whole exome sequencing authors found that Muc4 had frameshift indels in MMTV-PyVT mice. They mentioned the importance and role of Muc4 in tumor propagation in the discussion, but not checked in their samples, it would be interesting and important to check if there are any changes on their protein levels at different time points either by IHC or WB.

3.       For tumor, other than H&E staining and whole mount, nothing is done, it would be more logical if the expression profile of the molecular markers, Ki-67 was determined additionally either by IHC or western blot analysis with the tumors collected at different time points. Yes, it is done many times before, so does the use of this particular model.

4.       Because tumors were collected at different time points, this would be interesting to check at least one or two proliferative/anti or proapoptotic markers.

5.       In Discussion line-190-191, the sentence “Nonetheless, there is a significant difference in enrichment for gene sets between PyVT-induced tumors from the FVB and AKXD genetic backgrounds [18] is not necessary.

Author Response

Authors Chien-Liang Liu and colleagues describes the Characterization of mammary tumors arising from MMTV- 2 PyVT transgenic mice.  Authors analyzed whole-exome sequencing to identify constitutional and tumor-specific mutations, using the GRCm38/mm10 mouse reference genome. Tumors were collected at different time points for analysis including hematoxylin and eosin and whole-mount carmine alum staining. Authors found Frameshift insertions/deletions (indels) in the Muc4 and small indels and nonsynonymous single nucleotide variants but no somatic structural alterations or copy number variations. Though MMTV- 2 PyVT transgenic mice are used as mammary tumor model from long time, the current work author tries to find some new information by whole-exome sequencing.

In general, the manuscript is written well, importantly helpful for clinical and scientific understanding. There are few issues making the work less exciting and harnessing the readability. Overall, this manuscript is acceptable with the following minor comments:

For the tumors extracted at different time points mainly the authors did histology and whole-mount analysis. Mainly focus is Whole exome sequencing, and data analysis.

Point 1. It is not clear at what time point samples were collected for whole exome sequencing.

Response:

The samples were collected from mice at 12 weeks of age. We have added this information to the revised manuscript.

Point 2. Whole exome sequencing authors found that Muc4 had frameshift indels in MMTV-PyVT mice. They mentioned the importance and role of Muc4 in tumor propagation in the discussion, but not checked in their samples, it would be interesting and important to check if there are any changes on their protein levels at different time points either by IHC or WB.

Response:

We agree with the reviewer's comments and recognize that this is a limitation of the study.

Point 3. For tumor, other than H&E staining and whole mount, nothing is done, it would be more logical if the expression profile of the molecular markers, Ki-67 was determined additionally either by IHC or western blot analysis with the tumors collected at different time points. Yes, it is done many times before, so does the use of this particular model.

Response:

We thank the reviewer for suggesting that evaluation of the Ki-67 index may reinforce the results of histology and whole-mount analyses. We acknowledge that this is a limitation of the current study.

Point 4. Because tumors were collected at different time points, this would be interesting to check at least one or two proliferative/anti or proapoptotic markers.

Response:

We appreciate the reviewer for this insightful suggestion. Additional experiments were not performed because of time constraints related to the revision.

Point 5. In Discussion line-190-191, the sentence “Nonetheless, there is a significant difference in enrichment for gene sets between PyVT-induced tumors from the FVB and AKXD genetic backgrounds [18] is not necessary.

Response:

This sentence has been deleted from the revised manuscript.

Reviewer 3 Report

Comments and Suggestions for Authors

The study aimed to describe genetic and histological alterations during breast cancer progression. The manuscript is concise and generally well-written. However, unlike the histological assessment of the four distinct phases of cancer progression, as the authors have investigated, the lack of temporal assessment of genetic alterations at different phases of the tumor progression limits this reviewer's enthusiasm.

I would suggest the authors expand the depth of the current study by doing WES with samples collected at the remaining phases of the tumor progression that they have not investigated before. This would enable us to truly appreciate the genetic abnormalities occurring during tumor progression in this PyVT model and would possibly identify potential pathogenic mutations associated/responsible in a tumor-phasic manner.

Moreover, it was not mentioned precisely when the samples for WES were collected. The authors need to indicate this.

Author Response

The study aimed to describe genetic and histological alterations during breast cancer progression. The manuscript is concise and generally well-written. However, unlike the histological assessment of the four distinct phases of cancer progression, as the authors have investigated, the lack of temporal assessment of genetic alterations at different phases of the tumor progression limits this reviewer's enthusiasm.

I would suggest the authors expand the depth of the current study by doing WES with samples collected at the remaining phases of the tumor progression that they have not investigated before. This would enable us to truly appreciate the genetic abnormalities occurring during tumor progression in this PyVT model and would possibly identify potential pathogenic mutations associated/responsible in a tumor-phasic manner.

Response:

We thank the reviewer for this helpful suggestion. We did not perform WES at different time points because we were unsure that different somatic mutations could develop in a limited time frame. Nonetheless, it may be informative to carry out RNA-seq at different time points. We plan to include these in our future studies.

Comment:

Moreover, it was not mentioned precisely when the samples for WES were collected. The authors need to indicate this.

Response:

The samples were collected from mice at 12 weeks of age. We have added this information to the revised manuscript.

Round 2

Reviewer 3 Report

Comments and Suggestions for Authors

The authors had now added information about the age of the mice when the tumor samples were collected for WES.

I understand the authors' position regarding expanding the scope of the study. They hope to do RNA-seq in future studies. The current research can be published as is.